# Debittering of Grape Juice by Electrospun Nylon Nanofibrous Membranes: Impact of Filtration on Physicochemical, Functional, and Sensory Properties

**DOI:** 10.3390/polym15010192

**Published:** 2022-12-30

**Authors:** Maria Stella Cosio, Alessandro Pellicanò, Claudio Gardana, Carlos Alberto Fuenmayor

**Affiliations:** 1Department of Food, Environmental and Nutritional Sciences (DeFENS), University of Milan, Via Celoria 2, 20133 Milan, Italy; 2Instituto de Ciencia y Tecnología de Alimentos (ICTA), Universidad Nacional de Colombia, Carrera 30, Bogotá 111321, Colombia

**Keywords:** nanofibers, electrospinning, membrane filtration, bitterness, grapefruit juices

## Abstract

The effect of electrospun nylon-6 nanofibrous membranes (NFMs) on the concentration of bitter compounds and antioxidants of grapefruit juices during dead-end filtration processes was studied. Filtration experiments with aqueous standard solutions of different molecules showed that NFMs retain low molecular weight antioxidants (i.e., ascorbic and caffeic acids) only at early filtration stages, whereas they remove bitter glycosylated phenolics (i.e., naringin and narirutin) at a more stable ratio, variable according to the membrane thickness. Experiments with fresh grapefruit juice of two varieties (pink and yellow) showed that NFM-filtration reduces (17 to 30%) flavanones associated with the immediate bitterness and allows for the complete removal (>99.9%) of limonin, responsible for the persistent bitterness of many citrus juices. In contrast, the same process causes a lower loss of ascorbic acid (5%) and does not affect acidity, nor sugar concentration. The results confirmed that NFMs feature permselectivity towards bitterness-related compounds. This work highlights the NFM potential as filter devices for the selective reduction of the bitter terpenoid (limonin) and glycosylated flavonoids (naringin and narirutin) from grape juice citrus juices in the production of industrially-relevant beverages.

## 1. Introduction

Grape (*Citrus paradisi*) is a common commercial variety of *Citrus* belonging to the family of Rutaceae. Grapefruits are rich in bioactive compounds which have potential health benefits; but due to the bitter taste, their consumption is reduced. The bitter taste of grapefruits and juices is determined by the presence of flavonoids and terpenoids. Indeed, glycosylated flavanones, flavones, and flavonols are considered responsible for “immediate bitterness”, whereas terpenoids including limonoids (tetranortriterpenoid derivatives) are responsible for the “persistent” bitterness of juice [1,2,3,4]. In grapefruit, the flavanone naringin and the terpenoids as limonin have been identified as major bitter compounds [5,6,7,8]. From the perspective of sensory quality, typical concentrations of these compounds are usually considered acceptable, even distinctive, for grapefruit juices. However, the use of early-season fruits richer in naringin, the passage of limonin precursors (e.g., limonoate A-ring-lactone) from the peel into the juice due to squeezing, as well as chemical changes during industrial juice production, entail an increase in the concentration of bitter compounds. This often raises bitterness to less palatable levels representing a problem for the citrus industry [4,9].

A wide variety of technological approaches have been developed to reduce the excess of bitterness-related flavonoids and terpenoids from citrus juices and other products. Strategies such as (i) the modification of fruit handling protocols prior to the juice extraction (during harvesting or post-harvesting) [10,11]; enzyme-mediated degradation of compounds associated to the bitterness of the juice, in particular, flavonoids [2,12,13,14,15,16]; and physical processing using ionic exchange resins, polymers, or cyclodextrin inclusion complexes [2,5,17,18,19,20,21,22,23,24]. The adsorbent effectiveness varies, but removal rates of bitter compounds from citrus peel juices or molasses up to 80–90% have been reported [19].

Ultrafiltration in tandem with adsorptive techniques has also been reported to efficiently remove limonin from grapefruit juice [25]. However, microfiltration and ultrafiltration using membrane technology have been less explored in this context. Filter membranes are typically made of a polymer and produced by stretching melt-processed semi-crystalline polymer films, vapor or temperature-induced phase separation, irradiation, and extrusion [26]. Electrospinning is an alternative technique for producing polymeric membranes. It uses electric fields to spin polymer fibers with diameters from 100 to 10 nm, creating non-woven polymer mats [27]. The electrospun nanofibrous membranes (NFM) exhibit higher mechanical resistance, lower density, larger surface-to-mass ratio, and more convenient resistance-to-flow than their conventional counterparts [28,29,30,31]. For instance, Qin and Wang [28] compared the filtration performance of membranes constituted by non-woven electrospun nanofibers of poly(vinyl alcohol) (PVA, molecular weight M = 70 × 10^3^ g/mol with average fiber thickness of 200 nm) to those of polypropylene (PP) spun-bonded and melt-blown membranes. They found that by virtue of the very high surface area-to-volume ratio and sub-micron thickness of the electrospun membranes, not only their pore diameter was much smaller and homogeneous, but their filtration efficiency and resistance were higher. Moreover, electrospinning allows for tailoring pore size distribution and membrane thickness, which in addition to the polymer’s physicochemical nature and the possibility to modify its surface chemistry after membrane fabrication, have enabled improvements in the perm-selectivity of polymeric membranes for their use in tailored filtration [30]. In the context of their application in the production of food and beverages, several studies have demonstrated the potential of NFMs as filters to produce clarified juices and other beverages. Poly (ethylene terephthalate) (PET) nanofibers mats were successfully applied as filtration devices in apple juice clarification, obtaining juices similar to those clarified by ultrafiltration or by filtering aids, but with higher flux performances than the traditional processes [32]. Surface-modified electrospun polyacrylonitrile (PAN) nanofibrous membranes were used to selectively remove bromelain from aqueous media, with potential applications in the treatment of wastewater and pineapple juice production [33]. Polyamide (nylon-6) nanofibrous membranes have shown a potential for juice clarification and bacterial removal, demonstrating a superior performance than commercial membranes in the removal of turbidity, color, and bitter phenolic compounds, yet maintaining the antioxidant capacity of the juice [34,35,36]. Indeed, it has been reported that nylon-6 NFM have a selective adsorption capacity toward undissociated polyphenols (such as tannins) [34,37] which is a desirable feature for the bitterness reduction of beverages. 

The aim of this work was to evaluate the use of polyamide (nylon-6) NFM in dead-end filtration, as a non-thermal debittering process for juices of grapefruit (*Citrus paradisi*). The underlying hypothesis was that the filtration of fresh grapefruit juice using nylon-6 NFMs can reduce the concentration of compounds associated with the bitterness of this fruit, and therefore enables the perceived bitterness of the clarified product. First, the morphology of the nanofibrous membranes, as well as their chemical interaction behavior with bitter compounds, were studied. Subsequently, freshly obtained grapefruit juices of pink and yellow varieties and their corresponding NFM-filtrates, were analyzed and compared with commercial heat-treated grapefruit juices, in order to assess the impact of the filtration process on pH, acidity, soluble solids, color, total polyphenols content, phenolic profile, ascorbic acid, in vitro antioxidant activity and sensory profile.

## 2. Materials and Methods

### 2.1. Chemicals

All chemicals and solvents were of analytical reagent grade and were used without any further purification protocol. Sodium acetate trihydrate was purchased from Carlo Erba Reagents Srl (Milan, Italy). Folin Ciocalteau reagent, 2,2-diphenyl-1-picrylhydrazyl (DPPH⋅), sodium carbonate, acetic acid (glacial), formic acid, acetonitrile, ethanol, nylon-6, and analytical standards of narirutin, naringin, ascorbic acid, gallic acid, caffeic acid, glucose, fructose, sucrose, limonin, were purchased from Sigma Aldrich (Chemie Gmbh (Steinheim, Germany). All reagents used have a high purity (≥95%). 

Buffer acetate, pH 4.5 (0.1 M) was prepared from aqueous solutions of sodium acetate trihydrate and acetic acid. All the solutions were prepared using ultrapure water (18 MΩ cm) obtained with a Milli-Q Gradient A10, 0.22 μm filter (Millipore, Bedford, MA, USA).

### 2.2. Fruits and Juice Samples

Grapefruits (*Citrus paradisi*) from two different varieties: pink grapefruit of Israeli origin, and yellow (also known as white) grapefruit of Italian origin, were purchased from a local retailer and immediately processed. Grapefruit juices were obtained by directly squeezing the fruits with an model 67750 commercial and automatic extractor (Hamilton Beach, Glen Allen, VA, USA). The freshly squeezed juices obtained were centrifuged (4000 rpm, 10 min; 10 °C) in a centrifuge Universal 320R model (SilMar Instruments Srl, Milan, Italy) to remove excess pulp. Juices FJ-*p* (fresh juice from pink variety) and FJ-*y* (fresh juice from yellow variety) were then aliquoted according to their use, in samples for juice filtration experiments and samples for analyses. Filtration experiments were performed on the same day that juices were obtained.

The samples for physicochemical and chromatographic analyses were stored in several individual plastic tubes (50 mL) and kept for a few days at −20 °C until analysis; for each analysis, at least three tubes (replicates) were kept. All the fresh juices were analyzed unfiltered and filtered. 

Five different commercial heat-treated grapefruit juices, labeled as 100% natural, were purchased from local retailers and analyzed as comparison references. The first cluster of commercial juices consisted of three samples of UHT (ultra-high temperature)-treated grapefruit juices from grapefruit juice concentrate, of different commercial brands, packaged in 1000 mL Tetrapack^®^ containers and stored at room temperature, which were coded as HJ-1, HJ-2, and HJ-3. The second cluster consisted of two samples of pasteurized grapefruit juices (not from concentrate) of different commercial brands, packaged in 1000 mL Tetrapack^®^ containers and stored under refrigeration conditions, which were coded as HJ-4 and HJ-5. Except for the sample HJ-4, which declared yellow grapefruits as the single origin, there were no further indications available regarding the origin or variety of the fruits used for their manufacturing. After purchasing, the five commercial samples of juices (HJ-1, HJ-2, HJ-3, HJ-4, and HJ-5) were opened and immediately stored in several individual plastic tubes (50 mL) at −20 °C until analysis; for each physicochemical analysis, at least three tubes (replicates) were kept.

### 2.3. Commercial and Electrospun Nylon-6 Nanofibrous Membranes (NFM)

Whatman polyamide (nylon) syringe filters of 2.5 cm diameter in pre-built plastic holders with a nominal pore size of 0.45 μm (Whatman, Sprinfield, IL, USA) were used as a reference commercial membrane (CM). NFMs were prepared as described by Fuenmayor et al. [34]. Briefly, a 23% (*w*/*w*) solution of nylon-6 was prepared in formic acid. Plastic syringes (10 mL) fitted with a metallic needle (Hamilton) were filled with the polymeric solution and placed in a KDS 100 syringe pump (KD-Scientific, Inc., Holliston, MA, USA) at a flow rate of 0.15 mL/h. The needle of the syringe was connected to a Spellman SL150 high-voltage power supply by an alligator clip. A foil-covered copper tray, positioned at 11 cm in front of the needle and grounded, was used as a collector. For the electrospinning, the electrical potential was set at 25 kV. At the end of the electrospinning runs, the membranes were peeled-off. Membranes with different thicknesses were obtained by stopping the collection after different times (10, 20, and 40 min); the thickening ratio of the membranes during electrospun fibers deposition was in accordance with that described by Fuenmayor et al. [34]. Membranes were cut into circles of 2.5 cm diameter to fit dead-end filtration plastic holders.

### 2.4. Filtration Experiments

Dead-end filtration experiments of standard solutions of ascorbic acid (500 µg/mL), caffeic acid (10 µg/mL), naringin (50 µg/mL), and narirutin (5 µg/mL) in buffer acetate pH 4.5 (0.1 M), were (separately) done at a constant flow rate (10 mL/min) with NFM or CM, connected to 50 mL plastic syringes, using KDS 160 syringe pump (KD-Scientific, Inc., Holliston, MA, USA). Each 10 mL batch of the filtrate was collected separately, until completing 100 mL of filtered standard solution. The concentrations of each molecule were determined spectrophotometrically with a Cary 100 BIO UV-vis spectrophotometer (Varian, Inc., Palo Alto, CA, USA) in the standard solutions before filtration (*Cfeed*), and in each 10 mL batch sample (*Cfilt*), in order to evaluate the filtration capacity of the membrane for each specific compound. Concentrations were determined by measuring the absorbance of the solutions (directly for caffeic acid and narirutin, and 10X-diluted for ascorbic acid and naringin) at the characteristic absorbance peak wavelength (ascorbic acid: 254 nm; caffeic acid: 315 nm; naringin: 282 nm; narirutin: 282 nm) and correlating it with previously built calibration curves prepared in the same buffer. The change in the concentration, expressed as a percent filtrate-to-feed concentration ratio (*Cfilt/Cfeed*×100), was plotted against filtration time. 

For the juice dead-end filtration experiments, fresh juice samples, contained in 50 mL plastic tubes, were fed to the NFM or CM using an EMD Millipore peristaltic pump (Termo Fischer Scientific, Inc., Barrington, IL, USA) with 3 mm Nalgene^®^ tubings. The pump was set at maximum nominal speed, and the downstream flow rate quickly dropped: in a typical experiment, after 20 min the flow rate was 0.8 mL/min for the NFM obtained at 40 min of electrospinning collection time, 1.4 mL/min for the NFM obtained at 20 min of electrospinning collection time, and 0.6 mL/min for CM. Filtrates were subsequently collected in 30 mL plastic tubes, mixed after the completion of a filtration run, and kept in plastic tubes (50 mL) at −20 °C until analysis. At least three tubes (replicates) were kept for each physicochemical analysis. For the preparation of filtered juice samples for sensory analysis, several filtration runs were needed; a membrane was used for filtration runs of 30 mL each and replaced, and the filtrates of all runs were mixed and kept at +4 °C until analysis on the same day.

### 2.5. Chemical and Physical Analysis of Juice Samples

Grapefruit juices were characterized by measuring pH, total titratable acidity, total soluble solids, color, phenolic contents, and antioxidant activity. The pH was measured using a digital pH meter (pH M62, Standard Radiometer, Copenhagen, Denmark). Total titratable acidity was assessed on 10 mL of fruit juice diluted to 100 mL with distilled water, by titration with 0.1 M NaOH until reaching a pH 8.1; the values were expressed as % citric acid [38]. Total soluble solids were determined as °Brix by digital DBX-55 refractometer (Atago Co. Ltd., Tokyo, Japan). 

For the analysis of color, reflectance spectra were obtained by means of a Cary 100 Bio UV-vis spectrophotometer (Varian, Inc., Palo Alto, CA, USA). Blank measurements were made with ultrapure water and color coordinates were calculated as an average of the three replicates. The entire visible spectrum (380–770 nm) was recorded at 1 nm bandwidth. From the spectra, the color coordinates of the uniform color space CIELAB, (L*, a*, b*) and the Chroma [(*a**^2^ + *b**^2^)^1/2^] were calculated with Cary Win UV software (Varian), considering the Illuminant D65 and the 10° observer as references [39,40].

Total phenolic contents were determined spectrophometrically using Folin-Ciocalteu’s reagent according Singleton et al. [41]. Results were expressed as µg gallic acid equivalents (GAE) per mL of juices by comparison with a calibration curve built with the pure standard compound. 

Free radical scavenging activity was measured using DPPH radical as described previously by Brand-Williams et al. [42], with some modifications [43]. Different dilutions with ultrapure water of each juice were prepared and an aliquot of 0.1 mL was added to 3.9 mL of a 6.1 × 10^−5^ M DPPH⋅ methanolic solution and vortexed. The initial concentration (DPPH⋅) in the reaction medium was calculated by a calibration curve (Abs 515 = 11771x (DPPH⋅) + 0.01). The bleaching of DPPH⋅ was monitored at 515 nm by a model Uvidec-610 UV-visible spectrophotometer (Jasco International, Co., Ltd., Tokyo, Japan) and incubated at 30 °C for 30 min against a blank constituted by the juice solution containing all reagents except DPPH⋅ The scavenging activity of the samples on the DPPH⋅ was expressed as EC_50_ (μL of juice) and was extrapolated from a dose-response curve. All analyses were carried out in triplicate.

### 2.6. Chromatographic Analyses of Juice Samples

#### 2.6.1. Ascorbic Acid Analysis

Ascorbic acid (vitamin C) was determined by liquid chromatography with an HPLC system consisting of a Model 2080 plus PU pump and a UV-vis 2070 plus detector (Jasco, International, Co., Ltd., Tokyo, Japan), using a Fruit Quality Analysis column (100 × 7.8 mm, 5 μm particle size) (Bio-Rad, Laboratories, Inc., CA, USA). The chromatographic conditions were described by Mannino et al. [44]. The concentration was obtained by external calibration constructed in the range 1.25–10 µg/mL (p 95%, *n* = 10). The results were expressed as micrograms of ascorbic acid for mL of juice (µg/mL).

#### 2.6.2. Sugars Analysis

Sugars (glucose, fructose, and sucrose) were determined by HPLC equipment consisting of a model 880 PU pump (Jasco, International, Co., Ltd., Tokyo, Japan), with a model 400 thin layer electrochemical detector (EG&G, Princeton, NJ, USA), operating with a single glassy carbon electrode (surface area 8 mm^2^) at +500 mV, a reference (Ag/AgCl, satured) electrode and a platinum counter electrode. CarboPac PA 1 column (4 × 250 mm, 5 μm particle size) equipped with a CarboPac PA1 guard column (4 × 50 mm, 5 μm) (Dionex Corp., Sunnyvale, CA, USA) operating at room temperature. A detailed description of the system configuration has been reported elsewhere [45]. 

Calibration curves for glucose and fructose were constructed in the range of 10–50 μg/mL, for sucrose in the range of 50–250 μg/mL (*p* 95%, *n* = 5). The results were expressed as g/100mL. 

#### 2.6.3. Flavonoid Content Determination

Analysis of flavonoid composition was made using a mobile phase consisting of 70% ultrapure water with 0.1% formic acid (A), 30% acetonitrile (B), and 5 μL injected into a UHPLC system. Chromatographic conditions were described elsewhere [46].

Briefly, the analysis was performed on an Acquity UHPLC System (Waters Ges.m.b.H., Wien, Austria) coupled with an eLambda DAD (Waters) and a high-resolution Fourier transform Exactive Orbitrap mass spectrometer (Thermo Fischer Scientific, Inc., Barrington, IL, USA), equipped with a HESI-II probe for electrospray ionization (ESI) and a collision cell (HCD), using a BEH Shield C_18_ column (150 × 2.1 mm, 1.7 μm particle size; Waters). The MS data were processed using Xcalibur software (Thermo Fischer Scientific). The peak identity was ascertained by evaluating the accurate mass, the fragments obtained in the collision cell, and the online UV spectra (200–400 nm).

#### 2.6.4. Limonin Analysis 

Limonin was determined by an HPLC-UV [47]. HPLC analyses were performed at room temperature on an HPLC system consisting of a model 2080 plus PU pump and a UV–Vis 2070 plus detector (Jasco, International, Co., Ltd., Tokyo, Japan). A Kinetex, C_18_ Phenomenex (250 × 4.6 mm, 5 μm particle size) column was used. The injection volume and the detection wavelength were 5 μL and 210 nm, respectively. Working solutions of limonin in the range of 1–10 µg/mL were analysed. For the separation, a mobile phase consisting of water ultrapure and acetonitrile (65/35 *v/v*) in isocratic at a flow rate of 1 mL/ min was used. Integration of peak areas was performed with Borwin v. 1.2 software (Jmbs Developments, Grenoble, France). The results were expressed as micrograms of limonin for mL of juice (µg/mL).

### 2.7. Scanning Electron Microscopy

An electron scanning microscope SEM (Hitachi, Inc., Hillsboro, OR, USA) was used to observe the microstructure of NFM. Membranes, before and after 10 mL grapefruit (pink variety) juice filtration, were left overnight in a silica-drying chamber at room temperature. Specimens of approximately 4 mm × 4 mm were cut out and metallized with a 3–5 nm layer of Au coating. The specimens were bombarded with an electron beam at 30 kV in a vacuum. Images were obtained at magnifications of 4000× and 20,000×. 

### 2.8. Sensory Analysis

#### 2.8.1. Ranking Tests

A panel of thirty-two assessors performed two ranking tests [48] in 1 and 2 days, respectively. In the first test, FJ-*p* (fresh pink grapefruit of Israeli origin) and FJ-*y* (fresh yellow grapefruit of Italian origin) were compared to the three commercial UHT-treated juices, HJ-1, HJ-2, and HJ-3. In the second test, FJ-*p* and FJ-*y* were compared to the two commercial pasteurized juices, HJ-4 and HJ-5. The rank method allows for assessing differences among several samples based on the intensity of the attribute bitter. The samples were presented together in a randomized order, and the assessors were asked to rank for bitterness, using 1 for least bitter and 5 or 4 (for the first and second test, respectively) for most bitter.

#### 2.8.2. Paired Comparison Tests

The same panel of thirty-two assessors performed two paired comparison tests [49] in two separate sessions. Fresh grapefruit juices from yellow (FJ-*p*) and pink (FJ-*y*) varieties were compared to their corresponding NFM-filtrates using the thickest membrane or NFM-90 (FJ-*p* NMF-90, FJ-*y* NMF-90). Paired comparison test is a forced method that allows assessing differences of a specific attribute in a pair of samples. Assessors were informed about the procedure and purpose of the study and asked to indicate which of the two samples was the bitterest. Experiments were carried out in sensory panel booths under standardized conditions and under red light in order to mask even differences in color. The assessors were students and staff members of our department; all of them had modest to good experience in sensory evaluation. Mineral water was served between the samples to rinse the mouth.

### 2.9. Statistical Analysis

Mean values and standard deviations of all analyses were calculated for at least three independent replications. Data from physicochemical analyses were compared using a one-way analysis of the variance (ANOVA), and the differences between samples were evaluated by Tukey’s test at a 95% of confidence level. *p* < 0.05 was considered statistically significant. From sensory analyses, ranking data were analyzed through Friedman analysis [50] and paired comparison results were analyzed for significance by Student’s t-test. Statistical analysis was conducted by Statistica software 12 (StatSoft, Power Solutions Inc.; Palo Alto, CA, USA).

## 3. Results and Discussion

### 3.1. Filtration of Standard Solutions

Membranes of different thicknesses were prepared at collection times of 10, 20, and 40 min, corresponding to the thickness of approximately 20, 45, and 90 μm, respectively. These membranes were used to filter standard solutions of antioxidant species commonly found in citrus juices, either related (naringin, narirutine) or not (ascorbic acid, caffeic acid) to their bitterness. These filtration experiments were done at a constant flow rate (10 mL/min) using a syringe pump in a dead-end filtration arrangement. Each 10 mL batch of the filtrate was collected separately, until completing 100 mL of filtered standard solution. The concentrations before filtration (*Cfeed*) and in each 10 mL of filtrate (*Cfilt*) were determined in order to evaluate the membrane filtration capacity for each species.

Figure 1 shows the change in concentration, expressed as percent filtrate-to-feed concentration ratio (*Cfilt/Cfeed**100), plotted against filtration time, for ascorbic acid and caffeic acid. The plot corresponding to the change in vitamin C concentration shown in two stages (I and II, in Figure 1
*left*). At lower filtrate volumes (stage I), there was a noticeable increase in the antioxidant concentration in the filtrate. Membranes of different thicknesses behaved differently in this stage, with thicker NFMs showing larger retention capacities. At larger filtrate volumes (stage II), there was a nearly-constant vitamin C concentration in the filtrate, corresponding to a retention ratio of ~10–15%, regardless of the membrane thickness. Similarly, the plot of caffeic acid (Figure 1
*right*) evidenced two stages: an early stage in which caffeic acid concentration in the filtrate increased as filtration progressed, followed by a plateau value, i.e., a steady retention ratio. Thicker membranes achieved larger retention ratios of caffeic acid, as they did for ascorbic acid; however, for the latter, after the plateau value was reached (between 3–4 min, or 30–40 mL), the thicker membranes kept larger retention ratios: 14% for membranes 90 μm NFM (NFM-90) and 3% for 20 μm NFM (NFM-20). These results show that NFM can remove small-molecule antioxidants mainly at the early stages of filtration, after which the membranes appear to lose most of their removal capacity. Similar behavior has been reported about the NFM retention capacity of caffeic acid [34]. 

Figure 2 shows the change in concentration during filtration of bitter glycosylated phenolic compounds, naringin, and narirutin, using NFMs of different thicknesses and a nylon commercial membrane of known 0.45 μm pore size. In the plot depicting the change in naringin concentration (Figure 2
*left*), all the membranes showed a rather stable removal behavior, with a concentration in the filtrate increasing slowly. The NFM membranes presented retention capacities that were proportional to their thickness: at the final stage of the filtration experiments, retention ratios were around 5%, 20%, and 30% for thicknesses of NFM-20, NFM-45, and NFM-90, respectively. The commercial membranes had similar performance and their retention capacity was amongst those of NFM-45 and NFM-90. The results obtained with the standard solution of narirutin (Figure 2
*right*) were similar. The largest narirutin retention was achieved with the 90 μm-NFM, which retained 25–30% of the phenolic by the end of the filtration experiment.

The results of the preliminary filtration experiments with standard solutions indicate that NFM-90 can effectively retain molecules responsible for grapefruit bitterness to an extent that might be sensory-relevant. The membranes are less effective in retaining smaller antioxidant molecules such as ascorbic acid, which is a desirable feature since citrus fruits are expected to be good sources of vitamin C, and its content is an important freshness index.

### 3.2. Filtration of Grapefruit Juices

Freshly prepared juices of two varieties of grapefruit (FJ-*p* and FJ-*y*) were filtered using three types of membranes: NFMs with thicknesses of 45 μm (NFM-45) and 90 μm (NFM-90), and nylon commercial membranes (CM) of known 0.45 μm pore size. The 20 μm-NFM (NFM-20) was discarded for these experiments, due to its poor capacity to remove the tested bitter molecules.

Figure 3 shows the microstructure and macroscopic appearance of the nylon-6 NFM, before (Figure 3a) and after (Figure 3b) filtration of grapefruit juices, compared to those of commercial membranes (Figure 3c,d). Nylon-6 nanofibers presented a thickness of 95 ± 25 nm (*n* = 100) with the typical bead-free randomly oriented and interconnected arrangement of electrospun non-woven nanofibers, which results in membranes with pseudo-porous structures [32]. In comparison, CM exhibits the conventional porous surface of polyamide ultrafiltration membranes, responsible for their specific size-dependent separation capacity. After filtration, a layer of organic material can be observed on the surface of both NFM and CM. Although the fibrous microstructure of the NFM is still appreciable after a filtration run (Figure 3b), this severe material deposition, and the formation of an organic material layer, characteristic of dead-end membrane filtration processes, are signs of fouling, as reported by Gopal et al. [27] for polyvinylidene fluoride (PVDF) nanofibrous membranes. 

Physicochemical parameters (acidity, pH, °Brix, color), sugars, antioxidant components, and limonin content of the juices (FJ-*p* and FJ-*y*) and their corresponding filtrates, were assessed. These results were compared to those of commercial heat-treated grapefruit juices, either UHT-treated (HJ-1, HJ-2, and HJ-3) or pasteurized (HJ-4 and HJ-5), in order to evaluate the effect of the membranes on physicochemical quality, and bitter compounds removal during filtration. Table 1 shows the physicochemical parameters of grapefruit juices of two varieties, their filtrates obtained using three different membranes, and five varieties of commercial (heat-treated) grapefruit juices. All the samples presented titratable acidities between 1.1 and 2.4%, which are within [51], or slightly above [52], the expected range for grapefruit juices. Higher acidities were observed in fresh juices. These differences are attributable to different ripening degrees [53]. UH- treated juices (HJ-1, HJ-2, and HJ-3) presented acidities about 50% lower than those of the fresh juices, whereas pasteurized juices (HJ-4 and HJ-5) were among the two other groups. The values of pH were lower for fresh juices and higher for heat-treated juices, ranging between 2.9 and 3.5. There were no significant changes in either acidity or pH of the fresh juices attributable to the membrane filtration process.

Regarding soluble solids (Table 1), FJ-*y* presented higher values compared to those of FJ-*p*; however, when comparing all the samples together, only the UHT-treated HJ-1 had a significantly higher value (12.1 °Brix). There was a slight (although non-statistically significant) reduction in this parameter as a result of filtration: 3–4% for FJ-*p*, and 2–3% for FJ-*y*. The overall mean concentration of total assessed sugars (fructose, glucose, and sucrose) was around 7%, within the expected range of this parameter for grapefruit juice. The total sugars of the juice obtained from the yellow grapefruit and its corresponding filtrates were lower with respect to the other samples. There was not a significant reduction in sugars as an effect of the filtration process.

The values of color coordinates are presented in Table 1. There was a decrease in chromatic intensity or saturation (Chroma) due to the membrane filtration. The saturation reduction after NFM filtration varied between 20% and 55%, depending mostly on the membrane thickness. This decrease was more relevant when juices were filtered with thicker membranes (NFM-90) compared to those filtered with the 45 μm membrane (NFM-45), or the commercial membrane (CM). Lee et al. [20] observed a similar reduction, nearly 45% (from 10.2 to 5.6), of this parameter as a result of grapefruit juice debittering using XAD-16 resin.

Table 2 shows the concentration of total phenolic compounds, ascorbic acid, naringin, narirutin, and limonin, and an in vitro index of the antioxidant activity of the grapefruit juices and their filtrates, as well as those of the commercial grapefruit juices for comparison. Regarding the values of total phenolic compounds, the concentrations varied widely between the fresh juices of the two varieties, FJ-*p* and FJ-*y*. The mean concentration of phenolic compounds of FJ-*p* was 574 mg GAE/L. Filtering this juice with the NFM-90 and NFM-45 led to the removal of approximately 30% and 17%, respectively. Filtration with CM led to an intermediate reduction of nearly 20%. The mean concentration of phenolic compounds of FJ-*y* was 234 µg GAE/mL. Similarly, filtering FJ-*y* with NFM-90 and NFM-45 led to a reduction of approximately 30% and 18%, respectively, whereas in this case, the reduction caused by filtration with CM was more like that obtained with NFM-90 (~30%). In comparison, the commercial UHT-treated juices HJ-1 and HJ-2 presented the highest concentrations of total phenolic compounds amongst the samples analyzed (around 600 μg GAE/mL). The pasteurized juices, HJ-4 and HJ-5 presented concentrations of phenolic compounds more similar to the freshly prepared juices. In sum, nylon-6 membrane-filtered juices suffer a reduction in their total phenolic concentrations due to the filtration process, which is dependent on both the type and the thickness of the membrane. The removal of phenolic compounds was, in any case, lower than that reported by Lee et al. [20] using XAD-16 adsorption, which was 90% or higher. Compared to commercial samples, the membrane-filtered juices obtained from the FJ-*y* presented lower values of total phenolics, whereas those from the pink variety are among the values of heat-treated juices.

Regarding ascorbic acid (Table 2), FJ-*p* presented a concentration of 460 ± 14 μg/mL. In all FJ-*p* filtrates, there was a slight reduction of nearly 5%. The sample FJ-*y* had a concentration of 507 ± 11 μg/mL, with a similar extent of decrease in the corresponding filtrates. This agrees with the observations from the filtration experiments performed with standard solutions. In comparison, the loss of vitamin C reported by Lee et al. [20] was associated with resin-debittering was five times larger (26%). The ascorbic acid contents of the UHT-treated samples varied between 360 µg/mL and 434 μg/mL, whereas they were lower in the pasteurized samples. The concentrations of ascorbic acid found in FJ-*p* and FJ-*y*, agree with previous reports for these fruits, which vary from 280 –610 µg/mL [20,54,55].

Naringin represented approximately 80–90% of the total phenolic composition, while narirutin represented nearly 5% (Table 2). As an effect of the membrane filtration process, there was a reduction in the content of these two flavanones, which are partially responsible for the immediate bitter taste of grapefruit. The removal obtained with the different membranes roughly reflects the values obtained by filtration of standard solutions. With NFM-45, there was a reduction for both molecules of about 10–12%. With CM, the reduction was around 15%, while with NFM-90, the largest reduction was achieved, about 30%. Naringin and narirutin concentrations found in the samples of this study are between those reported by some authors [7,20,56] for grapefruit juices, while they were much lower than those reported by Xi et al. [52]. Apart from these two flavanones, only the aglycone naringenin was found in the chromatography analysis of the grapefruit juice samples in concentrations varying from 4 to 11 µg/mL, with no clear results regarding its reduction after membrane filtration. 

The antiradical capacity against free radical DPPH was determined, as an index of antioxidant activity. The result is expressed as EC_50_, that is, the amount of sample (μL) required to reduce the initial concentration of the radical by 50% in each amount of DPPH⋅ solution. The values of EC_50_ are 29.2 μL and 31.9 μL for FJ-*p* and FJ-*y*, respectively. For FJ-*p* this parameter varied from 29.2 μL to 33.3 μL after filtration with NFM-45, to 34.3 μL with CM, and to 36.2 μL with NFM-90. The trend is similar for the FJ-*y*. This indicates that filtration with nylon-6 membranes reduces to some extent the capacity of the juices of quenching free radicals, depending on the membrane morphology, due to the partial loss of antioxidant species, in particular phenolic compounds. In the rest of the juice, samples analyzed there was an antioxidant activity ranging from 24.4 μL to 30.6 μL in UHT specimens, while in the pasteurized samples from 34.3 to 36.6 μL. 

Interestingly, the most drastic changes in juice composition after filtration with the different membranes were related to their limonin concentration (Table 2). This compound is a major responsibility for the persistent bitterness of grapefruit juices. Concentrations of 99 ± 8 µg/mL of limonin were found in fresh juices of the pink variety, and of 83 ± 9 µg/mL in juices of the yellow variety. After filtration with NFM-45, the juices of the pink variety reduced their limonin content by nearly 97%, while for those of the yellow variety the reduction was roughly 95%. The commercial membranes, CM, allowed for a reduction of 99% for both juices. The greatest reduction of limonin content was achieved with the thickest nanofibrous membrane, NFM-90, which was >99.9% for both juices. The content of limonin in the commercial heat-treated samples of grapefruit juices varied from 59 ± 5 µg/mL (HJ-5) to 91 ± 9 µg/mL (HJ-4). Compared to the commercial samples or the unfiltered juices, any of the membrane-filtered grapefruit juices presented a much lower content of limonin.

The high selectivity of the membranes to remove bitter molecules is probably due to their relatively lower solubility in aqueous solutions, in the case of terpenoids, such as limonin. Moreover, the surface chemistry of the polyamide nylon-6 might be partially responsible for a greater affinity in the case of the glycosylated phenolics with higher molecular weight. Indeed, polyamide membranes are positively charged in acidic environments (pH < 4) with zeta potentials over 10 mV at the typical pH of grapefruit juices [57], which, in turn, could promote electrostatic affinity with the dissociated forms (oxoanions) of phenolic species, therefore contributing to perm-selectivity. 

### 3.3. Impact of NFM-Filtration on Bitterness

With the aim to investigate potential differences in bitterness among fresh grapefruit juices and commercial grapefruit juices two raking tests based on the attribute bitter were carried out. The ranking test 1 results (Figure 4A) showed a significant difference (*p* < 0.01) in bitterness among the five grapefruit juices samples, fresh (FJ-*p*, FJ-*y*) and UHT-treated (HJ-1, HJ-2, HJ-3). Fresh grapefruit juice yellow variety, FJ-*y*, and commercial HJ-1, were evaluated as the least bitter samples, followed by HJ-2. A significantly higher bitterness was perceived in the fresh grapefruit juice pink variety FJ-*p*. The bitterest sample was the HJ-3. The ranking test 2 results (Figure 4B) evidenced a significant difference (*P*<0.05) between the four grapefruit juice samples, fresh juices (FJ-*p*, FJ-*y*) and pasteurized juice (HJ-4, HJ-5). Fresh grapefruit juice yellow variety is still evaluated as the less bitter sample, together with commercial HJ-5. Significantly lower is the perceived bitter taste of fresh juice pink variety FJ-*p* and HJ-4. Both ranking tests evidenced that bitterness of the two varieties of fresh grapefruit juices was comparable with those of commercial UHT-treated and pasteurized products. In particular, the yellow variety was perceived as less bitter than the pink variety. These results agree with the lower content of limonin, naringin, and narirutin found in the yellow variety (Table 2).

With the aim to investigate the effect of filtration on bitter taste changing, two paired comparison tests were carried out. The numbers of assessors that perceived FJ-*p* and FJ-*y* after filtration with NFM-90 as less bitter than their correspondent fresh juices were major than the table minimum number to assess a significant difference (NFJ-*p*=; NFJ-*y*=; Ntable = 22). Results from both tests showed a significant (*p* < 0.05) reduction of bitterness after filtration with the nylon-6 nanofibrous membrane NFM-90.

## 4. Conclusions

The filtration experiments using standard solutions showed that NFM substantially removes small-molecule antioxidants, such as ascorbic acid and caffeic acid, only at the early stages of filtration, with retention ratios below 15%. The same membranes remove bitter glycosylated phenolic compounds, such as naringin and narirutin, at a more stable rate during filtration, with retention ratios as high as 30% for both compounds. Overall, the NFM capacity for removing these compounds depends on the membrane thickness, with thicker membranes exhibiting larger retention ratios, as expected. Filtration experiments using real samples of fresh grapefruit juice showed that NFM-filtration does not affect significantly the refractometric total soluble solids, nor juice acidity. On the other hand, NFM-filtration of grapefruit juice entails a chromatic intensity decrease, a slight loss of ascorbic acid (nearly 5%), and a reduction of approximately 17 to 30% of the concentration of total phenolic compounds, depending on the membrane thickness, as well as a similar reduction of the flavanones naringin and narirutin and of the DPPH radical scavenging capacity. These results agree with the standard solution filtration experiments.

NFM-filtration with the thickest membrane allowed for a nearly complete (>99.9%) removal of limonin. The physicochemical and functional characteristics of NFM-filtered grapefruit juices were consistent with the findings of the standard solution filtration experiments and confirms that nylon-6 NFM exhibit a higher degree of membrane permselectivity towards bitterness-related compounds. 

The selective removal of compounds that are responsible for the bitter taste of grapefruit juice, achieved by nylon-6 NFM filtration, reflected a reduction of the perceived bitterness as assessed by a panel of non-trained consumers. This confirms the hypothesis of the work and highlights the potential of NFMs as selective filters for debittering citrus juices and to produce clarified juices and other industrially relevant beverages.

## Figures and Tables

**Figure 1 polymers-15-00192-f001:**
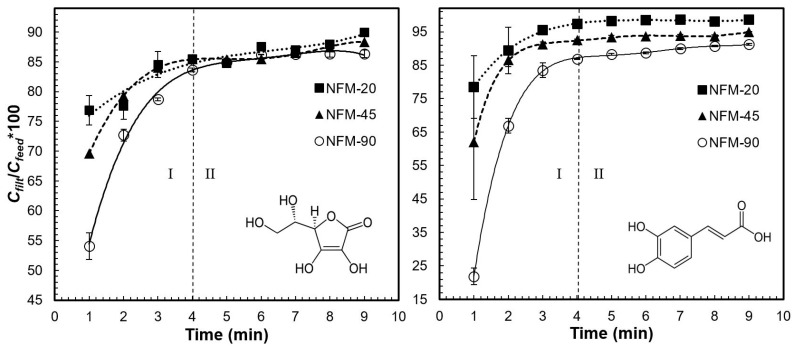
Change in concentration of standard solutions of typical small molecule antioxidant compounds: (*left*) ascorbic acid and (*right*) caffeic acid, during filtration with nylon-6 nanofibrous membranes of different thicknesses. Vertical error bars correspond to standard deviations of *n* = 3 repetitions.

**Figure 2 polymers-15-00192-f002:**
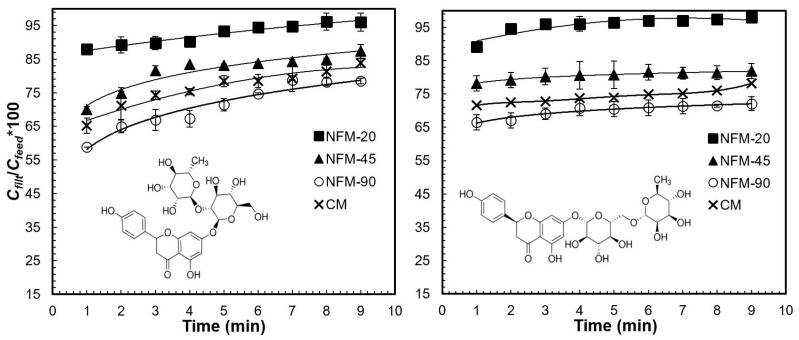
Change in concentration of standard solutions of typical *Citrus* glycosylated antioxidant compounds: (*left*) naringin and (*right*) narirutin during filtration with nylon-6 nanofibrous membranes of different thickness and nylon commercial membrane (CM). Vertical error bars correspond to standard deviations of *n* = 3 repetitions.

**Figure 3 polymers-15-00192-f003:**
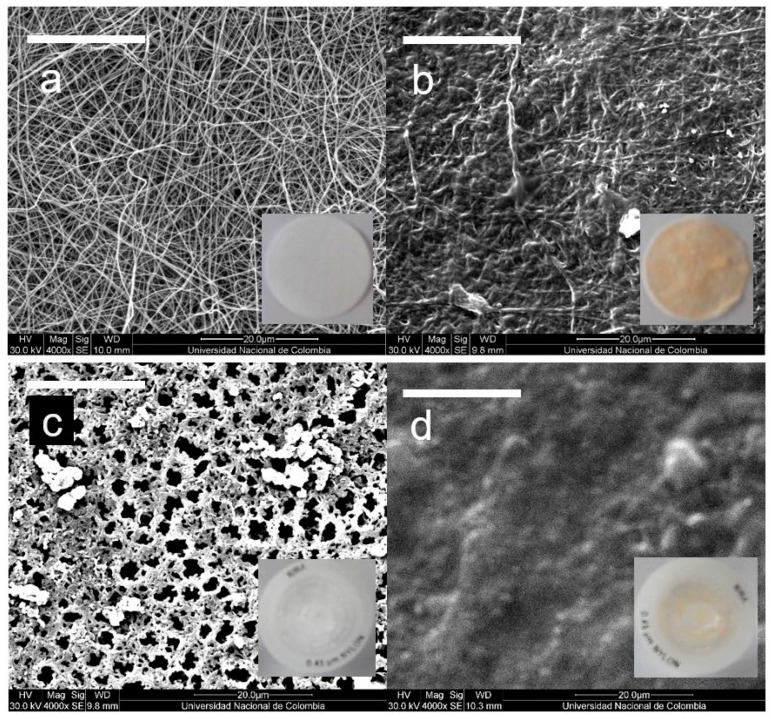
SEM images of the typical morphology of NFM (**a**) before and (**b**) after filtration, compared to the typical morphology of CM (**c**) before and (**d**) after filtration. In the inserts, the corresponding macroscopic appearance of the filters. Filtration conditions: 10 mL of grapefruit juice (pink variety). White bars correspond to 20 μm.

**Figure 4 polymers-15-00192-f004:**
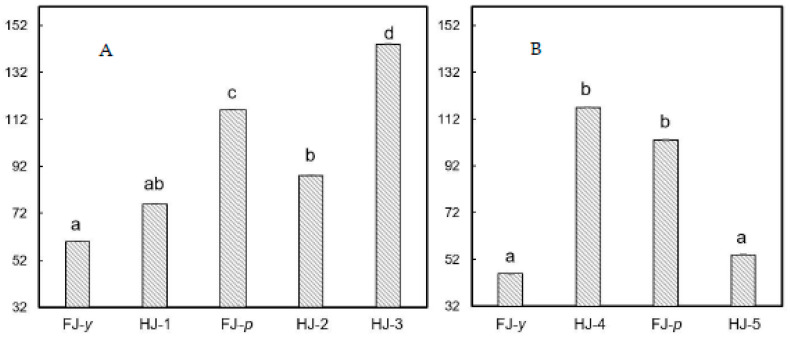
Ranking tests for bitterness: (**A**) among samples FJ-*p*, FJ-*y*, HJ-1, HJ-2, HJ-3; and (**B**) among samples FJ-*p*, FJ-*y*, HJ-4, HJ-5. Different letters indicate significant differences.

**Table 1 polymers-15-00192-t001:** Physicochemical parameters of grapefruit juices of two varieties, their filtrates obtained using nylon-6 nanofibrous membranes of different thicknesses (NFM-45, NFM-90), a nylon commercial membrane (CM), and five varieties of commercial (heat-treated) grapefruit juices.

Sample	Tritatable Acidity (g Citric Acid/100 mL)	pH	Brix	Fructose (g/100 mL)	Glucose (g/100 mL)	Sucrose (g/100 mL)	Total Sugars (g/100 mL)	Color			
								L*	a*	b*	Chroma
*Fresh grapefruit juices (pink variety)*											
FJ-1 (unfiltered)	2.4 ± 0.1 ^a^	2.9 ± 0.1 ^a^	9.8 ± 0.2 ^ad^	2.2 ± 0.2 ^a^	2.1 ± 0.1 ^a^	2.6 ± 0.2 ^a^	6.9 ± 0.2 ^ac^	98.5 ± 1.5 ^a^	−1.9 ± 0.2 ^a^	15.9 ± 1.1 ^a^	16.1 ± 1.4 ^a^
FJ-1 (NFM-45)	2.2 ± 0.1 ^a^	2.9 ± 0.1 ^a^	9.6 ± 0.1 ^ad^	2.2 ± 0.1 ^a^	2.1 ± 0.2 ^a^	2.5 ± 0.1 ^a^	6.8 ± 0.1 ^a^	96.7 ± 1.3 ^ab^	−1.4 ± 0.1 ^b^	12.9 ± 1.0 ^b^	12.9 ± 1.2 ^b^
FJ-1 (NFM-90)	2.2 ± 0.2 ^a^	2.9 ± 0.1 ^a^	9.6 ± 0.1 ^ad^	2.2 ± 0.2 ^a^	2.1 ± 0.2 ^a^	2.5 ± 0.1 ^a^	6.8 ± 0.1 ^a^	92.3 ± 1.3 ^c^	0.9 ± 0.1 ^c^	7.9 ± 0.4 ^c^	7.9 ±1.1 ^c^
FJ-1 (CM)	2.2 ± 0.1 ^a^	2.9 ± 0.1 ^a^	9.5 ± 0.1 ^a^	2.1 ± 0.1 ^a^	2.2 ± 0.2 ^ab^	2.4 ± 0.1 ^ab^	6.7 ± 0.2 ^ab^	93.5 ± 1.4 ^c^	−1.0 ± 0.2 ^c^	9.7 ± 0.6 ^d^	9.8 ± 0.8 ^d^
*Fresh grapefruit juices (yellow variety)*											
FJ-2 (unfiltered)	2.4 ±0.2 ^a^	3.0 ± 0.1 ^a^	10.6 ± 0.2 ^b^	2.2 ± 0.2 ^a^	2.1 ± 0.3 ^ab^	2.1 ± 0.2 ^ab^	6.3 ± 0.2 ^ab^	98.2 ± 1.3 ^a^	−2.6 ± 0.2 ^d^	19.0 ± 0.3 ^e^	19.2 ± 1.2 ^e^
FJ-2 (NFM-45)	2.3 ± 0.1 ^a^	3.0 ± 0.1 ^a^	10.5 ± 0.1 ^b^	2.2 ± 0.2 ^a^	2.1 ± 0.1 ^a^	2.1 ± 0.2 ^bb^	6.2 ± 0.3 ^ab^	98.4 ± 1.4 ^a^	−2.3 ± 0.2 ^d^	11.9 ± 0.4 ^b^	12.1 ± 1.3 ^b^
FJ-2 (NFM-90)	2.4 ± 0.1 ^a^	3.0 ± 0.1 ^a^	10.5 ± 0.1 ^b^	2.2 ± 0.2 ^a^	2.0 ± 0.2 ^a^	2.1 ± 0.2 ^ab^	6.3 ± 0.2 ^ab^	98.7 ± 1.5 ^a^	−1.9 ± 0.1 ^a^	8.4 ± 0.2 ^f^	8.6 ± 0.8 ^c^
FJ-2 (CM)	2.3 ± 0.1 ^a^	3.0 ± 0.1 ^a^	10.4 ± 0.1 ^b^	2.1 ± 0.2 ^a^	2.1 ± 0.2 ^a^	2.0 ± 0.1 ^b^	6.2 ± 0.2 ^b^	97.9 ± 1.6 ^a^	−2.0 ± 0.2 ^a^	9.8 ± 0.2 ^d^	9.9 ± 0.8 ^d^
*Commercial juices (UHT-treated)*											
HJ-1	1.1 ± 0.1 ^b^	3.5 ± 0.1 ^b^	12.1 ± 0.2 ^c^	2.5 ± 0.1 ^ab^	2.4 ± 0.1 ^ab^	2.3 ± 0.2 ^ab^	7.2 ± 0.1 ^ac^	75.4 ± 1.4 ^d^	3.2 ± 0.2 ^e^	29.8 ± 0.9 ^g^	29.9 ± 1.4 ^e^
HJ-2	1.3 ± 0.1 ^bc^	3.4 ± 0.1 ^b^	9.8 ± 0.1 ^ad^	2.8 ± 0.2 ^b^	2.6 ± 0.1 ^b^	1.8 ± 0.2 ^b^	7.2 ± 0.1 ^ac^	62.2 ± 1.3 ^e^	5.4 ± 0.2 ^f^	27.0 ± 1.3 ^h^	27.7 ± 1.5 ^f^
HJ-3	1.3 ± 0.1 ^bc^	3.4 ± 0.1 ^b^	9.9 ± 0.1 ^ad^	2.7 ± 0.1 ^b^	2.6 ± 0.2 ^ab^	2.0 ± 0.2 ^ab^	7.3 ± 0.2 ^ac^	85.6 ± 1.5 ^f^	0.9 ± 0.2 ^c^	23.0 ± 1.1 ^i^	23.0 ± 1.2 ^g^
*Commercial juices (Pasteurized)*											
HJ-4	1.6 ± 0.2 ^bc^	3.4 ± 0.1 ^b^	10.2 ± 0.2 ^ad^	2.6 ± 0.2 ^ab^	2.5 ± 0.1 ^ab^	2.1 ± 0.2 ^ab^	7.2 ± 0.2 ^ac^	83.5 ± 1.3 ^f^	1.1 ± 0.2 ^c^	22.3 ± 0.9 ^i^	22.9 ± 1.2 ^g^
HJ-5	1.8 ± 0.1 ^c^	3.3 ± 0.1 ^b^	9.8 ± 0.1 ^ad^	2.3 ± 0.2 ^ab^	2.2 ± 0.3 ^ab^	2.5 ± 0.2 ^ab^	7.0 ± 0.1 ^ac^	82.4 ± 1.5 ^f^	1.4 ± 0.2 ^b^	23.7 ± 1.0 ^i^	29.7 ± 1.4 ^e^

Results expressed as averages of *n* = 3 replicate ± the standard deviation. Different letters in the same column indicate significant differences.

**Table 2 polymers-15-00192-t002:** The concentration of total phenolic compounds, ascorbic acid, naringin, narirutin, and limonin, and an in vitro antioxidant activity (DPPH radical scavenging capacity) of two varieties of grapefruit juices, their filtrates obtained using nylon-6 nanofibrous membranes of different thickness (NFM-45, NFM-90), a nylon commercial membrane (CM), and five varieties of commercial (heat-treated) grapefruit juices.

Sample	Total Phenolic Compounds (µg Equiv. Gallic Acid/mL)	Ascorbic Acid (µg/mL)	Naringin (µg/mL)	Narirutin (µg/mL)	Antioxidant Activity (EC_50_)	Limonin (µg/mL)
*Fresh grapefruit juices (pink variety)*						
FJ-1 (unfiltered)	574 ± 10 ^a^	460 ± 14 ^a^	490 ± 14 ^a^	62 ± 6 ^a^	38.2 ± 2.5 ^a^	99 ± 8 ^a^
FJ-1 (NFM-45)	477 ± 17 ^b^	440 ± 12 ^a^	432 ± 12 ^b^	53 ± 2 ^b^	33.3 ± 1.8 ^ab^	3.3 ± 0.5 ^b^
FJ-1 (NFM-90)	424 ± 11 ^c^	435 ± 13 ^a^	343 ± 11 ^c^	42 ± 2 ^c^	36.2 ± 1.7 ^b^	0.04 ± 0.01 ^c^
FJ-1 (CM)	462 ± 15 ^b^	439 ± 15 ^a^	417 ± 10 ^d^	51 ± 2 ^b^	34.3 ± 1.4 ^ab^	0.89 ± 0.09 ^d^
*Fresh grapefruit juices (yellow variety)*						
FJ-2 (unfiltered)	234 ± 8 ^d^	507 ± 11 ^b^	175 ± 6 ^e^	38 ± 4 ^c^	37.3 ± 2.3 ^a^	83 ± 9 ^a^
FJ-2 (NFM-45)	190 ± 13 ^e^	489 ± 12 ^c^	154 ± 5 ^f^	34 ± 3 ^d^	32.7 ± 2.4 ^ab^	4.2 ± 0.8 ^b^
FJ-2 (NFM-90)	158 ± 17 ^f^	482 ± 13 ^ac^	120 ± 4 ^g^	25 ± 2 ^e^	36.4 ± 2.1 ^b^	0.04 ± 0.01 ^c^
FJ-2 (CM)	162 ± 9 ^f^	485 ± 11 ^ac^	149 ± 4 ^f^	33 ± 2 ^d^	34.0 ± 2.2 ^ab^	0.92 ± 0.07 ^d^
*Commercial juices (UHT-treated)*						
HJ-1	631 ± 15 ^g^	385 ± 20 ^d^	604 ± 15 ^h^	21 ± 4 ^e^	24.4 ± 2.5 ^c^	72 ± 11 ^e^
HJ-2	615 ± 7 ^h^	360 ± 11 ^e^	603 ± 13 ^h^	16 ± 3 ^e^	30.6 ± 3.5 ^a^	68 ± 8 ^e^
HJ-3	303 ± 9 ^i^	434 ± 18 ^a^	216 ± 8.0 ^i^	32 ± 4 ^d^	26.6 ± 1.5 ^c^	68 ± 7 ^e^
*Commercial juices (Pasteurized)*						
HJ-4	263 ± 11 ^l^	186 ± 19 ^f^	223 ± 9 ^i^	31 ± 2 ^d^	36.6 ± 3.6 ^b^	91 ± 9 ^a^
HJ-5	429 ± 10 ^c^	261 ± 11 ^g^	390 ± 8 ^l^	16 ± 1 ^e^	34.3 ± 1.7 ^ab^	59 ± 5 ^a^

Results expressed as the average of *n* = 3 replicate ± the standard deviation. Different letters in the same column indicate significant differences.

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
