# Peer review of "Debittering of Grape Juice by Electrospun Nylon Nanofibrous Membranes: Impact of Filtration on Physicochemical, Functional, and Sensory Properties"

_polymers, 2022, doi:10.3390/polym15010192_

Round 1

Reviewer 1 Report

The manuscript entitled 'Debittering of Grape Juice by Nanofibrous Membranes: Impact of Filtration on Physicochemical, Functional and Sensory Properties' is a very intersting article that uses novel memberane technology for debittering process without the use of chemical additives. Moreover, the quality of manuscript is very good. However I recocomend authors to go for english grammar and spell check. I have appended all the comments in attached PDF. 

Author Response

We have taken into account all the valuable comments made by the referees. We believe that the modifications introduced in the manuscript “Debittering of Grape Juice by Nanofibrous Membranes: Impact of Filtration on Physicochemical, Functional and Sensory Properties”, improved it considerably. As requested, you will find a point-by-point response. 

I have appended all the comments in attached PDF.

We submit the revised manuscript.

Reviewer 2 Report

Dear Authors, 

Your Manuscript presents an application of electrospun nylon-6 nanofibrous membranes (NFM) for filtration of grape juice. The English writing and presentation of the data were nice and I congratulate you for that. But, unfortunately, you focused on the food science side with no relevant work on the used polymer membrane (you changed only the thickness of the membrane). Since the readers of the journal are more interested in polymer sciences. I think that either the article has to be submitted to food science journal or at least add some more work on the used membrane for example:

i)                   Explain the electrospinning technique and condition and link them to the resulting fiber diameter and membrane porosity

ii)                 Discuss the effect of concentration PA6/HCOOH and polymer molecular weight on the structure and porosity of the prepared membrane, etc…).

iii)  The fiber diameter and the resulting porosity will be more relevant parameters for filtration so better to focus on how to change them.

Dear Authors, 

Your Manuscript presents an application of electrospun nylon-6 nanofibrous membranes (NFM) for filtration of grape juice. The English writing and presentation of the data were nice and I congratulate you for that. But, unfortunately, you focused on the food science side with no relevant work on the used polymer membrane (you changed only the thickness of the membrane). Since the readers of the journal are more interested in polymer sciences, I think that either the article has to be submitted to food science journal or at least add some more work on the used membrane for example:

i)                   Explain the electrospinning technique and condition and link them to the resulting fiber diameter and membrane porosity

ii)                 Discuss the effect of concentration PA6/HCOOH and polymer molecular weight on the structure and porosity of the prepared membrane, etc…).

iii) The fiber diameter and the resulting porosity will be more relevant parameters for filtration so better to focus on how to change them.

Best Regards

Author Response

Reference: Detailed response to reviewers

We have taken into account all the valuable comments made by the referees. We believe that the modifications introduced in the manuscript “Debittering of Grape Juice by Nanofibrous Membranes: Impact of Filtration on Physicochemical, Functional and Sensory Properties””, improved it considerably.

I have appended all the comments in attached PDF.

We submit the revised manuscript

Round 2

Reviewer 2 Report

Dear Authors,

The manuscript has been improved according to the previously submitted suggestions and the article in its present form fits the special issue for Applications in the Food Industry where you have been invited to publish your work.

Hence I recommend to publish this work in its present form and congratulate the authors for their work.

 Best Regards